# Prediction of Long Non-Coding RNAs Based on Deep Learning

**DOI:** 10.3390/genes10040273

**Published:** 2019-04-03

**Authors:** Xiu-Qin Liu, Bing-Xiu Li, Guan-Rong Zeng, Qiao-Yue Liu, Dong-Mei Ai

**Affiliations:** School of Mathematics and Physics, University of Science and Technology Beijing, Beijing 100083, China; grace41363011@126.com (B.-X.L.); chashaozgr@163.com (G.-R.Z.); qiaoyue_liu@163.com (Q.-Y.L.); aidongmei@ustb.edu.cn (D.-M.A.)

**Keywords:** deep learning, long non-coding RNAs, *k*-mer, BLSTM, CNN, GloVe

## Abstract

With the rapid development of high-throughput sequencing technology, a large number of transcript sequences have been discovered, and how to identify long non-coding RNAs (lncRNAs) from transcripts is a challenging task. The identification and inclusion of lncRNAs not only can more clearly help us to understand life activities themselves, but can also help humans further explore and study the disease at the molecular level. At present, the detection of lncRNAs mainly includes two forms of calculation and experiment. Due to the limitations of bio sequencing technology and ineluctable errors in sequencing processes, the detection effect of these methods is not very satisfactory. In this paper, we constructed a deep-learning model to effectively distinguish lncRNAs from mRNAs. We used *k*-mer embedding vectors obtained through training the GloVe algorithm as input features and set up the deep learning framework to include a bidirectional long short-term memory model (BLSTM) layer and a convolutional neural network (CNN) layer with three additional hidden layers. By testing our model, we have found that it obtained the best values of 97.9%, 96.4% and 99.0% in F1score, accuracy and auROC, respectively, which showed better classification performance than the traditional PLEK, CNCI and CPC methods for identifying lncRNAs. We hope that our model will provide effective help in distinguishing mature mRNAs from lncRNAs, and become a potential tool to help humans understand and detect the diseases associated with lncRNAs.

## 1. Introduction

Research shows that high-throughput sequencing technology has great power in profiling coverage and quantitative accuracy [1]. With the rapid development of high-throughput sequencing technology, a large number of transcripts have been found in many different species, including mammals such as humans and mice, and plants such as rice [2,3,4]. Long non-coding RNAs (lncRNAs) are sequences not encoding proteins with lengths greater than 200 nts, lacking or without an open reading coding frame [5]. With the development of research, lncRNAs previously regarded as “dark matter” or “garbage” have been gradually discovered; however, because of a lack of functional annotation, these RNAs have not been paid attention to for a long time. It was not until 2007 that Rinn et al. of Stanford University formally kicked off lncRNA research in an article published in *Cell* [6].

Studies have shown that lncRNAs play an important regulatory role in the processes of l life, and they are mainly involved in epigenetic regulation [7], transcriptional regulation [8] and post-transcriptional regulation [9]. lncRNAs are also involved in the development of various types of diseases, such as various cancers [10], leukemia [11], cardiovascular diseases [12], neurological diseases [13] and immune-mediated diseases [14].

Although there are some structural and functional differences between lncRNAs and mRNAs, they have similar PolyA cap structures. In addition, since high-throughput sequencing technology cannot guarantee the accuracy of the transcripts obtained, how to identify the two types of transcripts is also a difficult and challenging task.

With the discovery of non-coding RNAs, more and more biotechnologies are being used for detection. The technologies widely used in the study of lncRNAs are microarray, transcriptome sequencing technologies (RNA-seq), northern blot, real-time policy reverse transcription-polymerase chain reactions (qRT-PCRs), fluorescence in situ hybridization (FISH), RNA interference (RNAi) and RNA-binding protein immunoprecipitation (RIP). Microarray, with its ability to analyze global or parallel gene expression, can quickly assess differences in transcription profiles between different tissues or cell types, making it an ideal tool for finding targets [15]. Zhang et al. identified novel circulating lncRNAs in gastric cancer by using whole-gene lncRNA microarray analysis. The experimental results showed that five new plasma lncRNAs could be used as diagnostic biomarkers for gastric cancer detection [16]. Compared with DNA microarray, RNA-seq has a very low background signal because DNA sequences can be clearly mapped to specific regions of the genome, and RNA-seq has no quantitative upper limit, which is related to the number of sequences obtained [17]. Unlike hybridization, RNA-seq is based on deep sequencing and is not limited to detecting transcripts corresponding to known genome sequences. Because of its advantages, RNA-seq has been gradually adopted for transcriptome analysis. Although microarray and RNA-seq are excellent tools for finding targets, too many factors influence the final identification results from lab to lab, user to user and platform to platform. Northern blot can directly detect the presence of RNA transcripts and expression patterns in tissues, organs, developmental stages, environmental stress levels and therapeutic processes [18]. All these biocomputational detection methods have their own limitations such as high cost, complicated procedures and harm to the human body caused by the experimental process, which make them unavailable for universal promotion.

In recent years, due to the limitations of bio sequencing technology and ineluctable errors in the sequencing process, some predictive tools for lncRNAs and mRNAs have been developed. One of the common points of these tools is that machine learning is used to train the recognition model of lncRNAs and mRNAs. The Coding Potential Calculator (CPC) recognition algorithm has extracted the characteristics of lncRNAs and mRNAs, and used a support vector machine (SVM) for training to obtain the model [19]. The features of SVM model training by CPC come from two sources: one is to extract three features from the open reading frame (ORF) of the sample sequence, the other is to obtain three features from the homology of the protein sequence obtained by comparing the sample sequence with the protein library. Since CPC needs to be compared with the protein library, it is doomed to get good identification performance for the same species. Over the years, an algorithm called the Coding-Non-Coding Index (CNCI) has been proposed [20]. The overall framework of the CNCI algorithm contains two parts, one is the scoring matrix of the CNCI, which is related to different species with large calculation and poor portability, and the other is the classification model. CNCI can get high identification accuracy within a species. However, the recognition accuracy will be reduced when the sequence is mistaken, because the performance of this method depends on the quality of the sequences. PLEK is a recognition algorithm based on improved *k*-mer frequency as an input feature and SVM as classifier [21], which get an accuracy rate of over 90% in various datasets. Other methods such as the codon sub-situation frequency (CSF) algorithm and the PhyloCSF algorithm are developed based on the known protein library and the inherent characteristics of sequences, and they determine the category of the sequence based on the codon replacement frequency [22,23].

Most of these methods use machine learning based on surface learning, the main advantages of which are simplicity and convenience. However, due to the simplicity of surface learning, some complex characteristics of lncRNAs cannot be fully extracted, and the prediction performance cannot be further improved. In recent years, deep learning has been successfully used in a variety of biological fields, including genomics, transcriptomics, proteomics and structural biology, but minimal research has been done on the use of deep networks to identify lncRNAs. Fan et al. identified lncRNAs with an accuracy rate of 97.1% by combining multiple features of the open reading frame, *k*-mer, the secondary structure and the most-like coding domain sequence [24]. To accurately discover new lncRNAs, Pian et al. developed a random forest (RF) classification tool called lncRNApred based on a new hybrid feature. The mixed feature set includes three newly proposed features, namely MaxORF, RMaxORF and SNR, and the experimental results show better performance than CPC [25]. Yu et al. developed a double-layer deep neural network based on an auto-encoder of the hypothetical dataset, and achieved better a performance than traditional neural network [26]. Most methods use an open reading frame as the typical feature for distinguishing lncRNAs and mRNAs. In the absence of annotated information and sequence information, how to quickly and accurately identify lncRNAs from a large number of RNA sequences has become an urgent problem to be solved.

Min et al. developed a method to predict the chromatin accessibility via deep learning networks with *k*-mer embedding and got a good result [27]. Inspired by that article, we improved the model structure in [27] and used it to distinguish between lncRNAs and mRNAs. We outline the detailed model introduction in the Materials and Methods section.

## 2. Materials and Methods

It is critical to select a high-quality dataset with appropriate input characteristics for an accurate, fast and robust classifier. In this section, we will describe several common databases and data sources for training classification models, then give some explanations to the *k*-mer pattern used as classification feature. Finally, we will build the specific framework of the classification model. Experimental results and analytical discussions will be focused in Section 3.

### 2.1. Data Description

With the development of research, lncRNAs and other kinds of RNA molecules have been discovered, and some comprehensive transcriptional RNA databases have been produced. These have provided certain data sets for recognizing lncRNAs and mRNAs. RefSeq [28] is a database dominated and established by the National Center for Biotechnology in the United States. It contains comprehensive, integrated, non-redundant and well-annotated data, including mRNA sequences, lncRNA sequences, protein sequences and so on, as well as containing data on humans, mice and other species. The project ENCODE [29], led by the national human genome institute, aims to establish a complete list of functional components in the human genome. The GENCODE database [30] is annotated on the basis of the ENCODE database, including transcripts such as human and mouse mRNAs and lncRNAs. The Ensembl ncRNA database [31] contains non-coding RNA transcriptions of multiple species, related sequence information and functional annotations. NONCODE [32] is the most comprehensive database of non-coding RNAs established by the Chinese Academy of Sciences, which contains data of more than 1000 species. It is an open and comprehensive information platform. The lncRNAdb [33] provides information on lncRNAs of eukaryotes (including the source of lncRNAs, sequence information and functional annotation, etc.), which has high credibility. LNCipedia [34] is a database of human transcripts and genes with about 110,000 personal notes of lncRNA transcripts culled from different sources.

The RefSeq and GENCODE databases provide non-redundant and well-annotated sequence sets that can be used to build high-quality training and testing datasets.

### 2.2. Model Architecture

We used the *k*-mer pattern as the only classification feature in the mission of distinguishing lncRNAs from mRNAs. For each given RNA sequence with A, C, T and G nucleotides, we used a sliding window of length *k* and step *s* to intercept the sequence. Each subsequence with *k* nucleotides was called a *k*-mer pattern and all obtained *k*-mer patterns from an RNA sequence formed a *k*-mer sequence of length *L*. For given *k*, we can get 4k kinds of *k*-mer patterns. For example, when *k* = 2, we can get 16 kinds of patterns such as “AA”, “AC”, “AT” and so on. Next, we converted the *k*-mer patterns to numbers according to a dictionary in order to obtain the initial input features, where the dictionary meant giving each *k*-mer pattern an index like that “AA” corresponds to 1, and “AC” corresponding to 2. All the *k*-mer patterns finally corresponded to a set ℤ=[1,2,…,4k].

Each *k*-mer sequence, which corresponds to an RNA sequence, can be marked as “0” or “1”. “0” indicates lncRNAs and “1” indicates mRNAs. The aim in this paper is to construct a classifier to identify lncRNAs and mRNAs. The framework of the model is shown in Figure 1.

### 2.3. k-mer Embedding with GloVe

In the mission of identifying lncRNAs and mRNAs, features with global information must be more helpful. A traditional method based on *k*-mer pattern recognition, such as PLEK, only takes the frequency information into consideration but ignores the context information. Word2vec [35] is a classic model of word vector expression in natural language processes, but its disadvantage is that it is trained separately for each local context window without using the statistical information contained in the global co-occurrence matrix. Global vectors for word representation (GloVe) can consider both the global information and the statistical information contained in the global co-occurrence matrix [36]. Therefore, we chose the GloVe model for *k*-mer embedding. It can effectively extract statistical information by training on the nonzero elements in a word–word co-occurrence matrix. Figure 2 shows the entire embedding process.

In this paper, all samples were used to calculate a *k*-mer/*k*-mer co-occurrence counting matrix. We let the matrix of word–word co-occurrence counts be denoted by X, whose entries Xij tabulate the number of times word *j* occurs in the context of word *i*, where i,j∈[1,V] and V=4k. We defined that Xi=∑j=1NXij and Pi,k=Xi,kXi, Pi,k expressed the probability that the word *k* appears in the context of the word *i*, and found that the ratio of Pi,k and Pj,k had certain regularity. In cases where the word *j* was related to the word *k*, and the word *i* was related to the word *k*, this ratio approached 1; when they were not, the ratio was small. This regularity was the opposite when the word *j* and the word *k* were irrelevant. If we could use the word vector to calculate the ratio through some function to get the same regularity, it meant that our word vector had good consistency with the co-occurrence matrix, and also meant that our word vector contained the information from the co-occurrence matrix. According to the GloVe model, we get the embedded vector by training the loss function below
(1)J=∑i,j=1Xii≠0Vf(Xij)(wiTw˜j+bi+b˜j−logXij)2
where wi∈ℝD are embedding *k*-mer vectors that need to be obtained, w˜i∈ℝD are separate context *k*-mer vectors that are helping to get wi, bi are the biases for wi, b˜j are additional biases for w˜i to restore the symmetry and f(Xij) is a non-decreasing weight function containing two hyperparameters xmax and α. α is usually set to 34. We used the stochastic gradient descent method [37] to minimize the loss function in Equation (1) and obtain the embedded vector of all *k*-mers. At this point, we could complete the embedding stage through inputting x=[x1,x2,⋯,xL]∈ℤL and outputting [wx1,wx2,⋯,wxL]∈ℝD×L.

### 2.4. Bidirectional LSTM

In the mission of identifying lncRNAs and mRNAs, it is necessary to extract and process sequence information. The recurrent neural network (RNN) is a kind of neural network for processing sequence data. The biggest difference between the basic neural network and RNN is that the basic neural network only establishes the weight connection between the layers, while the RNN can not only achieve the weight connection between the layers but also establish the connection among units, which can better capture the connection between sequence data. Therefore, it is obvious that for each time step t, hidden layers $h_{t}$ are related to both the input unit $x_{t}$ in the time of t and the hidden layers in the last time step t−1.

Unfortunately, the method for training RNN parameters, called back-propagation through time, will lead to the problem "gradient vanishing" and "gradient exploding", which are caused by the cumulative multiplication of the derivative of the activation function. It is very difficult to learn RNN with long-term dependencies and gradient descent [38], so Hochreiter et al. proposed the long short-term memory (LSTM) network [39] to solve the problems mentioned above by introducing a memory unit, the key of which is the cell state. LSTM eliminated or added information to the cell state through a well-designed structure called a “gate”, and includes a sigmoid neural network layer and pointwise multiplication operation. 

Each LSTM unit, with an input vector ℝD in the time of t, consists of an input gate it, a control gate gt, a forget gate ft, a remember cell ct, an output gate ot and a hidden layer ht. The LSTM transition equations are
(2)it=σ(Wixt+Uiht−1+bi)
(3)gt=tanh(Wgxt+Ught−1+bg)
(4)ft=σ(Wfxt+Ufht−1+bf)
(5)ct=it⊙gt+ft⊙ct−1
(6)ot=σ(Woxt+Uoht−1+bo)
(7)ht=ot⊙tanh(ct)
where xt denotes the input in the time of t, W and U denote the weight metrics, b denotes bias, σ denotes logistic sigmoid function and ⊙ denotes pointwise multiplication. 

In our model, we output the sequence information at each moment to get a matrix that belonged to space ℝD*×L. To obtain a feature representation of the forward and backward information of the RNA sequence, we used the variant LSTM, a bidirectional LSTM (BLSTM), which consists of two parallel LSTMs: one input sequence forward and the other input sequence inverted.

### 2.5. Convolutional Neural Network

The convolutional neural network (CNN) was first used in 2D image processing [40]. Inspired by the biological natural visual cognitive mechanism, it is a special kind of multilayer feed forward neural network, and its characteristics are partial connection and parameter sharing. The handwriting character recognition model LeNet-5 with which we are most familiar is one of the most representative experiment systems in the early convolution neural network. The convolutional neural network has achieved great success in the field of digital image processing, and thus set off a frenzy of deep learning in the field of natural language processing. We applied it here to RNN sequence processing.

In the basic CNN model, each filter kernel has multiple two-dimensional feature maps to extract features of different directions. In natural language processing, we generally adopt the “single-layer CNN structure” [41]. There is not only one single layer, but a pair of convolution layer and pooling layer. Here, a one-dimensional kernel is used.

In the CNN model, the input layer is generally a number of matrices, and then is the convolution layer
(8)$$s(i,j)=(X*W)(i,j)=\sum_{m}\sum_{n}x(i+m,j+n)w(m,n)$$
where X represents the input matrix, W represents the convolution kernel and s(i,j) represents the corresponding position element value of the output matrix of the convolution kernel. The activation function of the convolution layer is usually a ReLU function as
(9)ReLU(x)=max(0,x)

The pooling layer is followed by the convolution layer. Compared with the convolution layer, the pooling layer is much simpler. Pooling is to compress each submatrix of the input tensor, keep the main features and reduce the parameter calculation to prevent overfitting. Pooling includes average-pooling and max-pooling, with no activation function. The hidden layer can be any combination of convolution layer and pooling layer. The most common combination is one convolution layer followed with one pooling layer. Finally, the full connection layer outputs a d-dimensional vector.

### 2.6. Prediction

At the end of the model, we regard binary classification as a logistic regression of feature representation in the stage of supervised training. Its advantage is that it directly models the probability of classification without presupposing the distribution of data, so it can not only predict the category, but also get the probability belonging to each category. Given the input xi and model parameter θ, the conditional probability of yi can be expressed as follows
(10)logp(yi|xi,θ)=yilogσ(βTci)+(1−yi)log(1−σ(βTci))
where β∈ℝd is the parameter to be predicted,ci∈ℝd is the d-dimensional vector for xi output from the convolution layer and σ(z)=1/(1+exp(−z)) is the sigmoid function. We can train the deep neural network by minimizing the following loss function
(11)l=−∑i=1Nlogp(yi|xi,θ)

The trained parameters are used to obtain the following classifier
(12)$$p(y_{i}=1|x_{i})=\frac{1}{1+e^{-\beta^{T}c_{i}}}$$
(13)$$p(y_{i}=0|x_{i})=1-\frac{1}{1+e^{-\beta^{T}c_{i}}}=\frac{1}{1+e^{\beta^{T}c_{i}}}$$

Then the classification of $x_{i}$ is determined by comparing $p(y_{i}=1|x_{i})$ and $p(y_{i}=0|x_{i})$.

## 3. Results and Discussion

To verify our model, we conducted a series of classification experiments using datasets collected from the GENCODE and RefSeq databases. First, in Section 3.1, we introduce the datasets prepared for classification and the details in the process of model training. Then, in Section 3.2, we compare the model with PLEK, CPC and CNCI. In order to evaluate the effectiveness of *k*-mer embedding as a feature, we also use Word2vec embedding in the model instead of GloVe, and compare it with the original model. In Section 3.3., we prove the effectiveness of the LSTM and convolution stages by proposing two variant depth-learning architectures. Finally, in Section 3.4, we perform a sensitivity analysis to show the robustness of our model.

### 3.1. Experimental Setup

To test the performance of our deep learning framework model, we chose two types of data: human and mouse. Human data included downloading human mature mRNA transcripts from the RefSeq database (version 90). After removing sequences less than 200 nts in length, 45,550 protein-coding transcripts were obtained in total. The human lncRNAs data was from the GENCODE database (version 28), containing 28,181 non-coding transcripts greater than 200 nts in length. We eliminated the dirty data less than 200 nts in length and any characters except “A”, “C”, “T” and “G”. In order to study the prediction ability of this model for cross-species, we also used the mature mRNAs of mice in the RefSeq database and the lncRNAs of mice in the GENCODE database. The specific data information is shown in Table 1, and the datasets of “human1” are attained from the article of PLEK. To ensure the balance of the dataset, all the data were used for experiment in a 1:1 ratio, then we randomly divided each type of data into strictly non-overlapping training and test sets with a ratio of 7:3. If the data was unbalanced, for the unbalanced category, we could not get the best results in real time, because the model would never fully investigate the implicit class. It also posed a problem for the acquisition of training and test samples, because in some cases where there were few observations, it was difficult to be representative in the class. The training set was used to train the neural network, and the test set was used to test the actual prediction ability of the model.

As for unsupervised training for *k*-mer embedding, we set *k* to 6 and the sliding window step size *s* to 6 to generate the corpus of *k*-mer sequences. With these parameters, we could get the *k*-mer for size $V=4^{6}=4096$. We implemented our own code for the co-occurrence matrix and trained GloVe with a package called mittens in Python. As for the hyperparameters of GloVe, the embedded vector dimension was set as 100, the window diameter of the co-occurrence matrix was set as 15, the cut-off value in the weight function was set as 15,000 and the maximum number of iterations was 3000.

We implemented our model through Keras, a deep learning library of Theano and TensorFlow, and chose TensorFlow as its backend, coded in Python. During the training process, we used the random gradient descent algorithm to optimize the cross-entropy loss function. The initial learning rate was 0.0001, and the batch size was set to 128. In order to prevent overfitting, we also adopted the early termination strategy, and the maximum number of iterations was set to 12.

### 3.2. Model Evaluation

First, we described the evaluation results of each dataset on our model via 10-fold cross-validation in Table 2 and listed the errors and accuracy in training and testing sets. It can be seen that the results of the training set and test set are very close, indicating that the early termination strategy we used effectively avoided overfitting. The best prediction was achieved on the human2 dataset, the accuracy of which reached 96.4%.

Next, we compared the performance of our model and several baseline methods, including PLEK (Aimin et al., 2014), CNCI (Liang et al., 2013) and CPC (Lei et al., 2007). For PLEK, we used Python to implement it by ourselves and we selected the set of hyperparameters mentioned by Aimin, with the best results to be recorded. For CPC, we directly found the website published in the article and tested it with our test dataset. For CNCI, we used the source code from the website https://github.com/www-bioinfo-org/CNCI and made it fit our dataset. For evaluation purposes, we used precision, recall, F1-score, accuracy, auROC and other evaluation indicators for the test set. The specific calculation formulas are as follows:(14)$$\mathrm{Pr}\mathrm{ecision}=\frac{TP}{TP+FP}$$
(15)$$\mathrm{Recall}=\frac{TP}{TP+FN}$$
(16)$$F1\mathrm{score}=2\times \frac{Precision\times Recall}{Precision+Recall}$$
(17)$$\mathrm{Accuracy}=\frac{TP+TN}{TP+TN+FP+FN}$$
(18)$$TruePositiveRate=\frac{TP}{TP+FN}$$
(19)$$FalsePositiveRate=\frac{FP}{FP+TN}$$
where FN stands for false negative, FP stands for false positive, TN stands for true negative and TP stands for true positive.

AuROC refers to the area under the curve of ROC. We took the “true positive rate” as the vertical axis and the “false positive rate” as the horizontal axis to draw the ROC curve, and calculated the area under the ROC curve (i.e., the value of auROC).

As can be seen from the comparison in Table 3 and Figure 3, our method always performed better than other baseline methods. Especially on F1score, accuracy and auROC, our model received better results than PLEK, CNCI and CPC. In human2 dataset, our model obtained the best values of 97.9%, 96.4% and 99.0% in F1score, accuracy and auROC, respectively, indicating that the deep learning of the automatic learning feature was more powerful than the support vector machine (SVM) feature of manually extracting the *k*-mer, and had a good prediction effect on cross-species detection. In addition, our method was always superior to the word2vec embedding method, and showed 0.043, 0.045 and 0.025 higher values in F1score, accuracy and auROC, respectively, than the word2vec embedding method. Although the precision of CPC was better than our model, the recall of it was too small, and under this circumstance it was proper for us to compare the F1score. Such an evaluation index showed the better performance of our method on the three datasets. In short, *k*-mer embedding showed better classification performance due to its contextual information contents. To show the stability of the model, we also used different random seeds to run our model multiple times and got the same results.

In order to implement the LSTM network, we used the zero supplement and truncation strategy, which is to fill the zero on the right side of the short sequence and truncate the long sequence to a maximum length set by us. Given the length distribution shown in Figure 4, we set the maximum length to 1000 in our experiment. For the sake of exploring the effect of this hyperparameter on the model, we set 500, 1000, 1500 and 2000 units, respectively, and then retrained our model on the human2 dataset. The test results are shown in Table 4. We found that classification performance decreased significantly when the length was 500 units because too much truncation resulted in the loss of most of the information. We also found that although the accuracy was the highest at 1500 units in length, the auROC was much lower than the length of 1000 units and the training time decreased as the maximum length decreased, so we tried to choose a smaller maximum length without affecting the performance of the model.

### 3.3. Efficacy of Deep Learning Network

A bidirectional LSTM model can effectively obtain long-term dependent information, which plays a crucial role in the prediction performance of the model. In our model, the BLSTM layer was applied, and the vector dimension of the neuron output at the last moment was set to 80. In order to confirm the validity of the BLSTM stage, we rebuilt a deep learning network variant with only an embedding layer and a convolutional layer, and the output of convolution layer was directly used as the final feature of classification. We tested this variant on the human dataset and recorded the final results in Table 5.

Table 5 recorded the evaluating indicators for the original model and the rebuilt model. As expected, the rebuilt models could not distinguish the mRNAs from lncRNAs, and regarded all the samples as mRNAs. Therefore, we can conclude that the BLSTM stage is essential in deep learning networks because it can process variable-length sequences and capture the long-term dependent information of the sequences.

In our model, the convolution phase consisted of three convolution layers, each layer of which had 100, 80 and 80 one-dimensional filter kernels with lengths of 10, 8 and 8, respectively, followed by a maximum pooling layer with lengths of 4, 2 and 2, respectively. Here, we designed a variant without the convolution phase to explore the validity of the convolution phase, then compared the results with the original model, and recorded the final results in Table 5.

By comparing the various indicators, we can find that the result is the same as that of the model without the BLSTM layer, which proves the importance of the convolution stage. Moreover, the maximum pooling layer in the convolution phase can greatly reduce the data dimensions and greatly reduce the computational complexity. Therefore, we consider it necessary to add the convolution phase to the model.

### 3.4. Sensitivity Analysis

To check the robustness of the model, we discussed the following three hyperparameters: the length *k* of *k*-mer, the embedding dimension *D* and the window sliding step *s* for sensitivity analysis. We tested on human2 datasets and used auROC as an evaluation indicator.

We valued *k* at 4, 5, 6 and 7, respectively. According to Figure 5a, we found that if the value of *k* was too small, not enough useful information could be extracted. If the value of *k* was too large, the exponential growth of vocabulary will occur. Therefore, we chose the value of *k* to be 6 for calculation.

Similarly, we took four different embedding dimension *D* values, including 50, 100, 150, 200. According to Figure 5b, we found that our model was not sensitive to *D* values. The range of auROC value changes was very small and could be ignored. We found that the larger *D* value meant that there were more parameters to be learned, which increased the complexity of model calculation and consumed more time; therefore, we chose the middle number 100 as the value of *D*.

In Figure 5c, we found that the value of auROC increased from 0.967 to 0.990 as the step length of the sliding window increases from 3 to 6, but declined over 6. This was because a large coverage area will result in insufficient information extraction, and the large size of *s* will reduce the size of the corpus, which will lead to the loss of information in the calculation of the co-occurrence matrix. For the above reasons, we chose *s* = 6 as the final sliding step length to make full use of the *k*-mer co-occurrence information.

## 4. Conclusions

In this paper, we proposed a deep learning network with pre-training *k*-mer embedding to differentiate lncRNAs and mRNAs by using only RNA sequences. We tested the model on two types of datasets, human and mouse, and compared it with PLEK, CPC and CICN. The results showed that the accuracy of our model is up to 96.4% and the auROC is up to 99.0%, which indicate that our model is better than all the methods above. The main contributions of this paper can be summarized as follows:

First, the unsupervised GloVe algorithm was introduced for the feature representation of RNA sequences, and the *k*-mer embedding vector was used instead of the traditional one-hot coding method. More contextual information and *k*-mer co-occurrence statistics were extracted, and a better feature representation was obtained.

Second, we constructed a new model using the BLSTM neural network first and convolution neural network second. The experiment result showed that the order of the two types of layers will affect the performance. The reason for this is that LSTM models tend to extract sequence information, and get access to long-term dependence between sequences, while a convolution model gets access to local information to extract more comprehensive information. If the convolution operation is carried out first, part of the sequence information from which LSTM should be extracted may be lost, and the classification performance may be limited.

In addition, compared with other baseline methods, our model expressed better classification performance. We confirmed the necessity of the BLSTM model and the CNN. Finally, the robustness of the model was demonstrated.

According to a series of discussion and experiments, the deep learning model with *k*-mer embedding has good classification performance and can better distinguish lncRNAs from mature mRNAs. We believe it could be a potential tool to help humans understand the detection of lncRNA-related diseases, which would help improve our understanding of the whole life process. Moreover, our research proved again that deep learning has great potential in the relative area.

## Figures and Tables

**Figure 1 genes-10-00273-f001:**
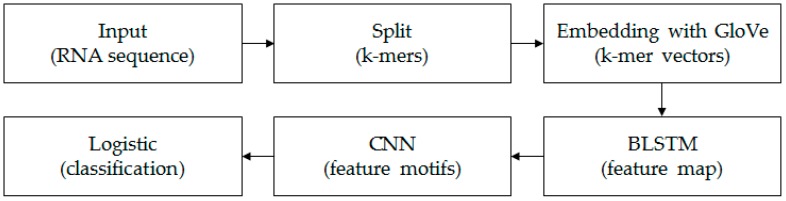
The structure diagram of the model. We first split each input RNA sequence into *k*-mers using a moving window approach [21]. Then, based on all *k*-mer sequences, all the *k*-mer embedding vectors were learned by the unsupervised GloVe method. The embedding layer embedded all *k*-mers into the vector space and turned the *k*-mer sequence into a real matrix. The BLSTM layer consisted of two LSTMs layers that were parallel but opposite in direction, to capture long-term dependency information between sequences. The following CNN with three convolution layers scanned the above results using multiple convolutional filters to obtain different features. The final fully connected layer and logistic acted as classifiers to get the probability and final classification result of the input sequence belonging to a positive or negative class.

**Figure 2 genes-10-00273-f002:**
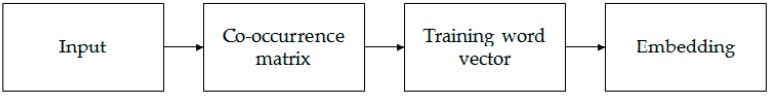
The process of embedding stage. Input all the *k*-mer sequences obtained in the previous stage, calculate the co-occurrence matrix for the entire *k*-mer corpus and get the embedded vector after training.

**Figure 3 genes-10-00273-f003:**
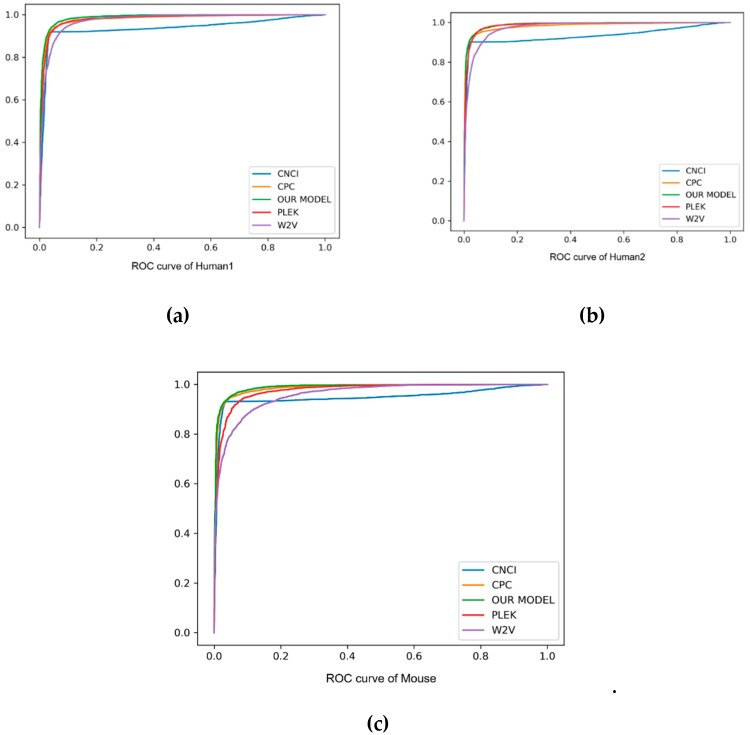
The ROC curve of human and mouse, with “false positive rate” as the horizontal axis and “true positive rate” as the vertical axis. Different method results are marked in different colors. Particularly, our method is marked in green.

**Figure 4 genes-10-00273-f004:**
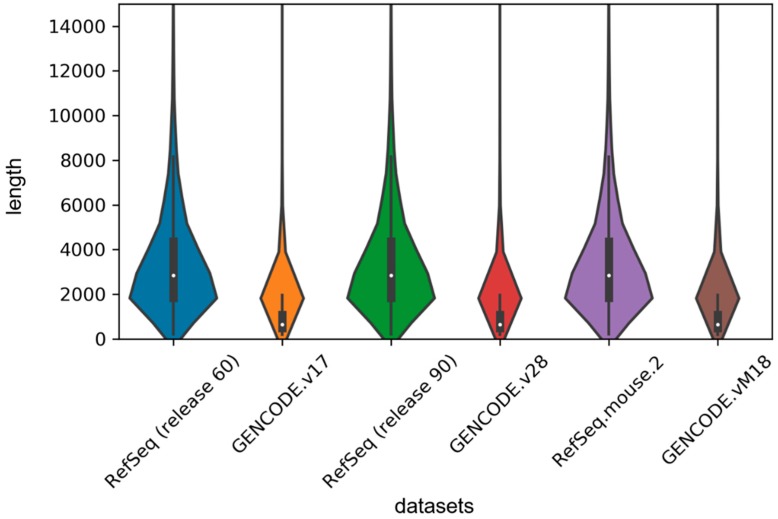
Violin plot for length distribution of RNA sequences in human and mouse datasets. The width of each violin indicates the size of dataset, and the white dots represent the median values of sequence lengths.

**Figure 5 genes-10-00273-f005:**
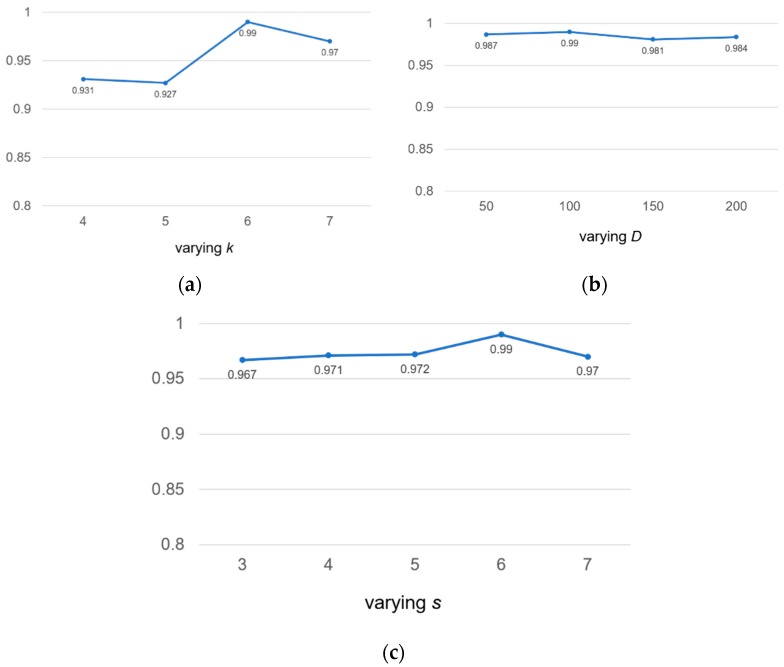
Sensitivity analysis of hyperparameters for *k*-mer length *k*, embedding dimension *D* and the stride *s* on the human2 dataset. auROC score are displayed.

**Table 1 genes-10-00273-t001:** Description of six datasets of mouse and human for lncRNA prediction.

Dataset	Database	Transcript	Size	Max Length	Min Length	Mean Length
Human1	RefSeq (version 60)	mRNA	22,389	109,224	201	3346
	GENCODE.v17	lncRNA	22,389	91,667	200	965
Human2	RefSeq (version 90)	mRNA	45,550	109,224	201	3346
	GENCODE.v28	lncRNA	28,181	205,012	200	1054
Mouse	RefSeq.mouse.2	mRNA	15,896	24,271	224	3208
	GENCODE.vM18	lncRNA	17,624	93,147	200	1404

Database denotes the source of our article data, which came from the RefSeq and GENCODE databases. Size denotes the number of sequences the dataset contained, and max, min and mean length denote the maximum, minimum and mean values of sequence lengths for each dataset in nts, respectively. Note that we deleted any sequence shorter than 200 nts from the lncRNA dataset.

**Table 2 genes-10-00273-t002:** Detailed results of our model on each dataset, including cross-entropy loss and accuracy on training and test datasets.

Dataset	Train Loss	Test Loss	Train Accuracy	Test Accuracy
Human1	0.143	0.167	0.966	0.959
Human2	0.137	0.154	0.973	**0.964**
Mouse	0.166	0.175	0.953	0.949

**Table 3 genes-10-00273-t003:** Classification performance for four different methods in lncRNA prediction experiments.

Dataset	Tool	Precision	Recall	F1score	Accuracy	auROC
Human1	PLEK	0.950	0.968	0.959	0.949	0.987
	CNCI	0.962	0.919	0.940	0.938	0.936
	CPC	0.975	0.849	0.908	0.913	0.978
	Word2vec	0.897	**0.978**	0.936	0.917	0.969
	**Our Method**	**0.982**	0.971	**0.976**	**0.959**	**0.988**
Human2	PLEK	0.951	0.965	0.958	0.949	0.987
	CNCI	0.954	0.901	0.927	0.953	0.932
	CPC	**0.993**	0.836	0.908	0.897	0.982
	Word2vec	0.900	**0.976**	0.936	0.919	0.965
	**Our Method**	0.982	**0.976**	**0.979**	**0.964**	**0.990**
Mouse	PLEK	0.930	0.919	0.925	0.929	0.976
	CNCI	0.957	0.931	0.944	**0.949**	0.947
	CPC	**0.983**	0.838	0.905	0.917	**0.984**
	Word2vec	0.885	0.884	0.884	0.891	0.956
	**Our Method**	0.943	**0.980**	**0.961**	**0.949**	**0.984**

This table records the various indicators for evaluating the performance of the model, including precision, recall, F1score, accuracy and auROC value. Best results are shown in bold.

**Table 4 genes-10-00273-t004:** Our model performance on the human2 dataset with different maximum length of input sequences on the BLSTM stage.

Length (Units)	Precision	Recall	F1score	Accuracy	auROC
2000	0.975	0.982	0.979	0.963	0.988
1500	0.962	0.989	0.975	0.982	0.957
1000	0.982	0.976	0.979	0.964	0.990
500	0.968	0.953	0.960	0.932	0.977

The indicator values of precision, recall, F1score, accuracy and auROC under input sequences with different lengths separately.

**Table 5 genes-10-00273-t005:** Classification performance of two variant deep learning architectures and our original model.

	Precision	Recall	F1score	Accuracy	auROC
Full	0.982	0.976	0.979	0.964	0.990
No BLSTM	0.861	1	0.925	0.861	0.746
No conv	0.861	1	0.925	0.861	0.746

“Full” means the original model, including an embedding stage, a BLSTM stage and a convolution stage; “No BLSTM” means the variant architecture removing the BLSTM stage; “No conv” means the variant architecture removing the convolution layer.

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
