# Peer review of "Prediction of Long Non-Coding RNAs Based on Deep Learning"

_genes, 2019, doi:10.3390/genes10040273_

Round 1
Reviewer 1 Report
Liu et al. report on a method of long non-coding RNAs prediction based on deep learning.
Comments:
Line 3 – Title- Learning misspelled
A more thorough description of the Glove algorithm is needed.
Details of the supervised deep learning framework including the bidirectional long short-term memory model (BLSTM) need to be added.
What criteria are used to show that the new model shows better classification performance than the traditional PLEK, CNCI, CPC models.
Line 150- describe moving window approach (add reference).
Line 260- How were the datasets from the GENCODE and refSeq databases chosen?
Minor spelling, grammatical and syntax errors throughout the text.
Author Response
Dear Reviewer:
I am very happy that you can read my article carefully and thank you very much for giving so many good suggestions. I will respond to these questions you mentioned in the comments point-by-point. Thank you again for your review.
Point 1: Line 3 – Title- Learning misspelled.
Response 1: The title spelling error has been corrected in the article. Thanks for your reminder.
Point 2: A more thorough description of the Glove algorithm is needed.
Response 2: More detailed explanations have been added in section 2.3(In lines 175-183). Thanks for your suggestion.
Point 3: Details of the supervised deep learning framework including the bidirectional long short-term memory model (BLSTM) need to be added.
Response 3: The Deep Learning Framework has been presented in the article, and more details can be found in Ref. 39 and 40. As for the bidirectional long short-term memory model (BLSTM), the specific LSTM process is given in section 2.4(In lines 202-220). BLSTM is to input the feature vector twice on the basis of LSTM: one input sequence forward and the other input sequence inverted. The specific parameter settings are mentioned in section 3.3(In lines 381-386, 397-401). Thanks for your suggestion and we hope we have made a satisfactory answer.
Point 4: What criteria are used to show that the new model shows better classification performance than the traditional PLEK, CNCI, CPC models.
Response 4: In our article, the criteria used to show our model different from other models are precision, recall, f1score, accuracy and auROC. For example, our model obtained the best value of 97.9%, 96.4% and 99.0% in F1score, accuracy and auROC respectively. After considering these criteria comprehensively on three data sets, we found that our model is better than other models we mentioned. The more specific analysis is mentioned in section 3.2.
Point 5: Line 150 – describe moving window approach (add reference).
Response 5: The moving window approach is used to select k-mer patterns. For each RNA sequence, we use a sliding window of length k, step s to intercept the sequence to obtain subsequences with k nucleotides. Reference has been added to the moving window approach and the specific method descriptions are mentioned in section 2.2(In lines 136-140).
Point 6: Line 260 – How were the datasets from the GENCODE and refSeq databases chosen.
Response 6: These two databases are commonly used in study of such problems. The data we select are all with good annotation information. We downloaded mature mRNA transcripts as positive class from the RefSeq database (version 90) and the lncRNA sequences as negative class from GENCODE database (version 28) by removing sequences less than 200nts in length. The specific introduction is mentioned in section 3.1(In lines 276-285). We hope this reply will satisfy you.
Point 7: Minor spelling, grammatical and syntax errors throughout the text.
Response 7: We have checked the spelling, grammatical and syntax errors of the article and corrected them. Thanks for your suggestion.
In addition, according to another reviewer’s suggestion that article title is a little bit general, we have revised the title as follows: Prediction of Long Non-coding RNAs Based on Deep Learning.
Thank you very much!

Reviewer 2 Report
Dear authors,
I have read your article, entitled "Study on the Method of Long Non-coding RNAs Prediction Based on Deep Learning", where you introduce a methodology to distinguish mature mRNAs from lncRNAs.
In my opinion the article is clear to read, and the topic tackled is, without any doubt, of great interest. The introduction of the work is well written, as you recall the related state of the art and it is clearly introduced the scope of the problem. The explanation of the model is clear and complete. I have few remarks and some questions that I would like you take into account for the final version of the article.
In my opinion the title is not too much explanatory, and to me seems a little bit general, at first glance. I suggest you to change it, making it a little more specific.
The abstract is too long. In rows 12-20 it is briefly explained the model, but without clearly mention about the structure of it. In my opinion, in this form these lines are useless for the reader. I suggest to reduce the length of these rows, and to write something more regarding lines 9-11, and 21-24.
In line 277, what do you mean by writing "To ensure the balance of the data set, all the data were used for experiment in a 1:1 ratio". I suggest you to explain better this choice, because it is crucial for understanding the reported results.
why have you selected k = 6 for k-mer embedding? Have you tried many other values? The same question holds for hyper parameters selection of GloVe.
In line 278 you write that "we randomly divided each type of data into strictly non-overlapping training set and test set, with a ratio of 7:3.", but in line 300 you write that you performed a 10-fold cross validation. Then, to me it is not clear what you have done. Have you selected the best parameters with a 10-fold cross validation on training test, and then tested the model on test set? Please explain me.
Author Response
Dear Reviewer:
I am very happy that you can read my article carefully and thank you very much for giving so many good suggestions. I will respond to these questions you mentioned in the comments point-by-point. Thank you again for your review.
Point 1: In my opinion the title is not too much explanatory, and to me seems a little bit general at first glance. I suggest you to change it, making it a little more specific.
Response 1: Thank you for making this suggestion. We also think this title is a little bit general. According to your suggestion, we have revised the title as follows: Prediction of Long Non-coding RNAs Based on Deep Learning. We hope this will be a satisfactory answer.
Point 2: The abstract is too long. In rows 12-20 it is briefly explained the model, but without clearly mention about the structure of it. In my opinion, in this form these lines are useless for the reader. I suggest to reduce the length of these rows, and to write something more regarding lines 9-11, and 21-24.
Response 2: The abstract has been reworked in the article, reducing the description of the method and focusing more on the background and experimental results. Thanks for your suggestion.
Point 3: In line 277, what do you mean by writing "To ensure the balance of the data set, all the data were used for experiment in a 1:1 ratio". I suggest you to explain better this choice, because it is crucial for understanding the reported results.
Response 3: I have explained the reasons for choosing balanced data in section 3.1(lines 287-290), thanks for your suggestion and we hope we have made a satisfactory answer.
Point 4: why have you selected k = 6 for k-mer embedding? Have you tried many other values? The same question holds for hyper parameters selection of GloVe.
Response 4: In section 3.4, we have explained these issues. For k, we valued k at 4,5,6 and 7, respectively. When we selected k=6, we got the best result. For the D and s of GloVe, we took four different embedding dimension D values, including 50,100,150,200 and selected the s=3,4,5,6,7. When we selected the middle number 100 as the value of D and chose s=6 as the final sliding step length, the model showed the best classification performance. In section 3.4(In lines 408-423), we have explained the reasons in detail and showed the results in figure 5 visually.
Point 5: In line 278 you write that "we randomly divided each type of data into strictly non-overlapping training set and test set, with a ratio of 7:3.", but in line 300 you write that you performed a 10-fold cross validation. Then, to me it is not clear what you have done. Have you selected the best parameters with a 10-fold cross validation on training test, and then tested the model on test set? Please explain me.
Response 5: We are sorry that we have not explained clearly about what you confused in the article. As you said, we randomly divided each type of data into strictly non-overlapping training set and test set, with a ratio of 7:3. In the training set, we used 10-fold cross-validation to select the best parameters, and then used the test set to evaluate the classification performance of the model after obtaining the final model. We hope this reply will satisfy you.
Thank you very much!
